# Characterization of Lymphocyte Subsets in Lymph Node and Spleen Sections in Fatal Pediatric Malaria

**DOI:** 10.3390/pathogens11080851

**Published:** 2022-07-28

**Authors:** Wilson L. Mandala, Steve Ward, Terrie E. Taylor, Samuel C. Wassmer

**Affiliations:** 1Academy of Medical Sciences, Malawi University of Science and Technology, Thyolo 310106, Malawi; 2Malawi Liverpool Wellcome Trust Clinical Research Programme, Blantyre 312233, Malawi; sam.wassmer@lshtm.ac.uk; 3Department of Tropical Disease Biology, Liverpool School of Tropical Medicine, Liverpool L3 5QA, UK; steve.ward@lstmed.ac.uk; 4Blantyre Malaria Project, Kamuzu University of Health Sciences, Blantyre 312233, Malawi; ttmalawi@msu.edu; 5College of Osteopathic Medicine, Michigan State University, E. Lansing, MI 48824, USA; 6Department of Infection Biology, London School of Hygiene & Tropical Medicine, London WC1E 7HT, UK

**Keywords:** *P. falciparum* malaria, lymphocytes, lymph nodes, spleen

## Abstract

Secondary lymphoid tissues play a major role in the human immune response to *P. falciparum* infection. Previous studies have shown that acute falciparum malaria is associated with marked perturbations of the cellular immune system characterized by lowered frequency and absolute number of circulating T cell subsets. A temporary relocation of T cells, possibly by infiltration to secondary lymphoid tissue, or their permanent loss through apoptosis, are two proposed explanations for this observation. We conducted the present study to determine the phenotype of lymphocyte subsets that accumulate in the lymph node and spleen during acute stages of falciparum malaria infection in Malawian children, and to test the hypothesis that lymphocytes are relocated to lymphoid tissues during acute infection. We stained tissue sections from children who had died of the two common clinical forms of severe malaria in Malawi, namely severe malarial anemia (SMA, n = 1) and cerebral malaria (CM, n = 3), and used tissue sections from pediatric patients who had died of non-malaria sepsis (n = 2) as controls. Both lymph node and spleen tissue (red pulp) sections from CM patients had higher percentages of T cells (CD4^+^ and CD8^+^) compared to the SMA patient. In the latter, we observed a higher percentage of CD20^+^ B cells in the lymph nodes compared to CM patients, whereas the opposite was observed in the spleen. Both lymph node and spleen sections from CM patients had increased percentages of CD69^+^ and CD45RO^+^ cells compared to tissue sections from the SMA patient. These results support the hypothesis that the relocation of lymphocytes to spleen and lymph node may contribute to the pan-lymphopenia observed in acute CM.

## 1. Introduction

Although significant gains have been attained in reducing case numbers and mortality over the past decade [1,2] global malaria burden still remains high, with *Plasmodium falciparum* malaria accounting for an estimated 241 million clinical cases and 627,000 deaths in 2020 [3]. Falciparum malaria can manifest either as uncomplicated (UM) or severe infection. The latter includes cerebral malaria (CM) and severe malarial anemia (SMA) as well as other complications, with a number of possible overlaps [4]. CM is associated with the worst outcome, contributing to the highest number of deaths [5], and to a wide spectrum of neurological sequelae in survivors [6]. Although the pathogenesis of the different clinical types of severe malaria is poorly understood, they result from a combination of host and parasite factors, including high inflammation and sequestration of infected red blood cells (iRBCs) onto the vascular endothelium and a subsequent inflammatory response [7], leading to end-organ damage.

Natural protective immunity against malaria takes years to develop, even in high transmission areas and despite repeated exposure to the parasite. This immunity is not only parasite-specific [8], but also stage-specific [9,10], and is influenced by age, genetics, pregnancy, nutritional status and co-infections [9]. A reduction in exposure can lead to rapid waning of this acquired immunity, potentially increasing the risk of developing more severe forms of the disease [10]. 

Although vaccines have proven to be the most reliable, cost-effective and efficient method for controlling the burden and spread of many infectious diseases, this has not been the case with malaria. The most promising malaria vaccine candidate to date, RTS,S/AS01, was approved for pilot implementation programme in three African countries in 2015 [11], Malawi being one of the three. The search for a robust and effective malaria vaccine continues and a better understanding of naturally acquired immune responses to the various stages, including the transmissible stages of the parasite, could be crucial in informing effective vaccine designs.

The spleen plays an important role in immunity against malaria, through the mediation of humoral and cellular immune responses, and by physically clearing altered host erythrocytes [12]. Various leucocyte subsets migrate through different zones of the spleen, therefore increasing the chance of detecting *P. falciparum* antigens [13]. Antigen-presenting cells (APCs) such as monocytes, macrophages and dendritic cells (DCs) phagocytize iRBCs and parasite debris in the marginal zone before migrating to the white pulp, where they activate naïve and memory T cells [14].

Studies in murine models of malaria have shown a depletion of T and B cells from the marginal zone of the spleen with a corresponding increase of the same cells in in the red pulp during the course of a *P. chabaudi chabaudi* malaria infection [15,16]. This observation was corroborated by a postmortem study of adult Vietnamese patients who died of *P. falciparum* malaria [17]. Spleen sections in this cohort showed an altered architecture with a distinct dissolution of the splenic marginal zones and substantial loss of B cells [17]. 

Lymph nodes are also important in the host immune response against malaria, as T and B cells migrate to different segments, where they interact with antigen-carrying APCs and undergo clonal expansion [18,19]. The proliferating cells create germinal centers of activation, proliferation, differentiation and death of B lymphocytes [20]. 

Our group [21] and others [22] have previously shown that acute falciparum malaria is associated with marked perturbations of the cellular immune system characterized by lowered frequencies and absolute numbers of T cells in the peripheral circulation. A temporary relocation of T cells, possibly by infiltration to secondary lymphoid tissue, and permanent loss of these cells through apoptosis are the two leading explanations suggested for this observation [23,24]. 

While other groups explored this phenomenon in pediatric UM [23] and in acute *P. falciparum* or *P. vivax* malaria [24], we conducted the present study to (i) investigate whether lymphocytes relocate to and accumulate within lymphoid tissues in Malawian pediatric patients with severe *P. falciparum* malaria (either CM or SMA); and (ii) characterize the phenotypes in both lymph nodes and spleen. We used archived samples from children who died of CM or SMA collected as part of a postmortem study at Queen Elizabeth Central Hospital (QECH) in Blantyre, Malawi [25,26]. Samples from patients who died of non-malarial sepsis were used as controls. 

## 2. Results

### 2.1. Antibody Titration Results

Tissue labelling using CD56, a marker of natural killer (NK) cells, did not result in any distinguishable positive cells that could clearly be identified as NK cells in any of the three patient groups. Similar negative results were obtained with different antibody concentrations (no dilution, 1:25, 1:50, 1:100 and 1:200) and incubation times (1 h, 2 h, 12 h, 24 h and 48 h). Consequently, we excluded this marker from all subsequent analyses. Staining with CD69 and CD4 only resulted in good outcome when the incubation was done for 24 h; the other primary antibodies resulted in good staining after 1hr incubation (Table 1).

### 2.2. Lymphocyte Phenotyping in Lymph Node Sections

Lymph node tissue sections from CM patients (Figure 1) had a similar percentage of T cells to those observed in sections of control patients who died from non-malaria complications. In contrast, the SMA patient had lower percentage of T cells in lymph node sections compared to both CM and control patients. Lymph node tissue sections from CM and SMA patients had similar percentages of CD4^+^ T cells, which were marginally lower compared to control tissues. CM patients had the highest, and those of SMA patients had the lowest percentage of CD8^+^ T cells in their lymph nodes of the three groups. Tissue sections from SMA patients had the highest percentage of CD20^+^ B cells (Figure 1 and Figure 2). 

Patients who died of non-malaria related complications had higher percentages of CD69^+^ and CD45RO^+^ cells in their lymph node sections compared to the percentage of these cells in tissue sections from CM and SMA patients (Figure 1). CM patients had a higher percentage of these two cell types compared to the levels observed in SMA.

### 2.3. Lymphocyte Phenotyping in Spleen Sections

Spleen tissue sections from the control patients had a higher percentage of T cells compared to CM and SMA patients, which showed similar levels across both groups (Figure 3 and Figure 4). CM patients had a higher percentage of CD4^+^ and CD8^+^ T cells in their spleen sections compared to controls and SMA patients, and the last group had the lowest percentage of these cells (Figure 3). Tissue sections from CM patients also had a higher percentage of CD20^+^ B cells and CD69^+^ cells compared to the other two groups. SMA patients had the lowest percentage of CD69^+^ and CD45RO^+^ cells of the three groups (Figure 3).

## 3. Discussion

Pan-lymphopenia is a hallmark of severe *P. falciparum* malaria [21,22,23,24]. This phenomenon is not well understood but may result from a temporary, disease-induced relocation of lymphocytes, especially CD4^+^ and CD8^+^ T cells, to secondary lymphoid tissue [24]. Upon treatment and cure, the pan-lymphopenia reverses and a marked increase in lymphocyte counts is observed in the peripheral circulation [21,22]. Here, we show that lymph node and spleen tissue sections from CM patients had higher percentages of CD4^+^ and CD8_+_ T cells compared to a patient with SMA; the percentage of B cells in their spleen sections was also higher. In contrast, the SMA patient had a higher percentage of CD20^+^ B cells in lymph node sections compared to CM patients. The percentages of CD69^+^ (a marker of early activation) and CD45RO^+^ (a marker of memory T cells) were higher in lymph node and spleen sections from CM cases than in the SMA patient.

The observation that lymph node and spleen tissue sections from CM patients had higher percentages of CD4^+^ and CD8^+^ T cells compared to a patient with SMA might somewhat seem counter-intuitive, since T cells have previously been linked with immunopathology observed in patients presenting with CM [23]. This apparent paradox may be partly explained by the fact that only 5% of lymphocytes are present in peripheral circulation suggesting that various lymphocyte subsets may translocate to secondary lymphoid compartments such as spleen and lymph nodes where they encounter antigens for their subsequent maturation and differentiation. Another possibility is that lymphocyte subsets are lost through apoptosis, which would result in permanent cell loss. The rapid normalization of the observed lymphopenia and our histologic findings seem to support the temporary translocation hypothesis. Lymphoid cells sequestering in the secondary lymphoid tissues of CM patients rapidly rejoin the peripheral circulation once the infection is cleared or controlled.

To date, there is only one study that describes patterns of architectural reorganization in the spleen from adult patients who died from severe malaria in Vietnam [17]. As differences in immunity [27] and clinical presentation [28] have been described between Asian adults and African children with severe falciparum malaria, our study aimed to assess whether similar cellular processes occurred in children from Malawi. Indeed, while the sequestration of iRBCs in secondary lymphoid tissues has been investigated using postmortem tissues [25,26], similar studies aimed at characterizing migration and transient relocation of different leucocyte subsets in secondary lymphoid tissues during malaria infection have mainly been conducted in murine models [16,29,30]. 

Results of early murine studies showed marked decreases in the numbers of circulating lymphocytes in peripheral circulation 2 to 4 days after infection [28,31]. A similar trend was reported in human infection [22,23]. These numbers then returned to control levels 6 to 8 days post-infection. Subsequent cell trapping experiments showed that peripheral blood lymphocytes were preferentially recruited in the spleen during the initial stages of the infection with Th1 lymphocytes as the main cell type affected [32]. 

We demonstrate a high percentage of CD4^+^ and CD8^+^ T cells in splenic and lymph node sections of CM patients, which is consistent with the reported cellular alterations in the spleen during *P. berghei* infection [29]. Indeed, the numbers of CD4^+^ and CD8^+^ T cells in the spleen of resistant mice strains increase rapidly during infection and are maintained at a high level before migrating into the brain before the onset of experimental CM [29].

In adult patients with severe malaria, the overall number of T cells in the red pulp increased significantly, and there was a profound depletion of B cells from the marginal zone surrounding B cell follicles [17]. Here, we report for the first time similar findings in the spleen and the lymph nodes in fatal pediatric CM, but not SMA. Additional analyses are warranted to confirm potential differences between age groups during fatal SMA. Worth noting is the observation that the lymph node sections from SMA case had on average less than 10% CD3^+^ (total T) cells as compared to more than 20% CD4^+^ T cells. This anomaly could have arisen from the possibility that a higher proportion of the T cells were actually CD4^+^ T cells with very low proportion of CD8^+^ T cells as indicated in Figure 1. 

In addition, the investigators of the Vietnamese study also found that the frequency of germinal centres within lymphoid follicles was significantly reduced in severe malaria, indicating that the differentiation of B cells into plasma and memory cells was compromised [17]. This is consistent with our results, as spleen sections from CM patients had lower percentages of CD20^+^ B cells compared to the other groups.

In a study on the phenotype and localization of B cells in mice models during infection with *P. chabaudi chabaudi*, investigators showed that there was a depletion of B cells from the marginal zone and profound disruption of the T-cell areas. Similar to the SMA case we present here, intensive germinal centre formation and extra-follicular B cell proliferation were observed in the red pulp [30].

The predominance of CD69^+^ cells in the spleen of CM patients is consistent with the results of the Vietnamese study [17], where the expression of the activation marker HLA-DR on sinusoidal lining cells was increased in the red pulp in fatal malaria cases. This too is consistent with the results of our previous work [21], in which severe malaria cases were characterized by highly activated T cells in periphery as shown by a high expression of the activation marker CD69.

In a separate study to determine lymphocyte migration in murine malaria infected with *P. chabaudi chabaudi* [16] investigators found that although the white pulp of the spleen in infected mice increased in size 11 days after infection, there was a partial lymphocyte depletion which coincided with the occurrence of massive lymphocytosis in the peripheral blood. The investigators also showed a decreased uptake and retention of T and B cells by the spleen of infected mice but an increased retention of T and B cells in the liver and lungs of infected mice [16]. The fact that lymphocyte traffic to the spleen was found to be reduced in murine studies [16,30] but higher numbers of T and B cells were observed both in our cohort and in the Vietnamese one [17] suggests that the relocation of lymphocytes to the spleen could be a feature specific to human CM. 

The reduced lymphocyte traffic to the spleen in infected mice coincided with an increased lymphocyte retention in the peripheral circulation, lungs and especially in the liver [16]. This is consistent with the lymphocytosis we described in acute SMA [21]. Immunohistochemical analysis of liver tissue sections from patients may, in the future, provide additional information on lymphocyte trafficking in human SMA. Indeed, studies assessing the importance of the increase in T and B cells traffic to the liver in malaria infection have also provided evidence for a crucial role of the accumulation of leucocytes in the liver to the development of protective immunity during malarial infection [32]. Thus, it is possible that reduced entry of lymphocytes into the spleen may in fact contribute to increase the delivery of committed immunologically competent cells to the liver, where parasite destruction occurs [32].

Our study had several limitations, including a small sample size due to the low availability of these specific tissues in our cohort. In addition, although the staining for some cells such as CD4^+^ and CD8^+^ T cells and B cells was not as ideal in the malaria cases, we were still able to distinguish between stained and unstained cells. However, we were unable to optimize the staining for NK cells despite numerous variations of our staining protocol in terms of incubation time and dilution of antibodies. Nevertheless, our findings in pediatric CM are in line with previous ones from Vietnamese adults and contribute to a better understanding of the relocation of lymphocytes to secondary lymphoid organs during *P. falciparum* infection across different age and geographical groups. Only cervical lymph nodes were used in this study, so additional analyses are warranted to assess whether the proposed malaria-induced relocation of lymphocytes we report also occurs in lymph nodes from other parts of the body. Further studies including liver tissue sections should be considered to investigate the accumulation of leucocytes in the liver, as previously shown in murine malaria [32]. It is also possible that proliferation of tissue-resident cells could have contributed to the lymphocyte counts observed in the tissue sections from CM cases, a hypothesis that should be explored in future work. 

Overall, our results are consistent with the hypothesis that the relocation of lymphocytes to spleen and lymph node contribute to the pan-lymphopenia observed in acute CM. 

## 4. Materials and Methods

### 4.1. Postmortem Tissue Sections

Postmortem spleen and lymph node tissue sections were obtained from the Blantyre Malaria Project Autopsy Study. Cervical lymph node and spleen samples were collected from children who died of severe malaria on the Pediatric Research Ward at Queen Elizabeth Central Hospital (QECH) in Blantyre, Malawi from 1996 to 2009 [25,26]. Sections from three patients diagnosed with CM and one patient with SMA were analyzed for this study, as well as additional samples from two controls cases who died of non-malaria sepsis were analyzed (Table 2).

### 4.2. Ethical Clearance

The original study was approved by the IRB at Michigan State University and by the College of Medicine Research and Ethics Committee (COMREC) of the former College of Medicine of the University of Malawi, now Kamuzu University of Health Sciences (KUHeS). Informed consent was granted by the families of the deceased, and all agreed to the future use of stored samples in studies of malarial pathogenesis.

### 4.3. Immunohistochemical Analysis of Post-Mortem Tissue Sections

#### 4.3.1. Reagents

Tris-buffered saline (TBS) was prepared by mixing 6.1 g of Trizma base (Sigma, Darmstadt, Germany) with 9.0 g of sodium chloride (Sigma) in 800 mL of distilled water. The pH was adjusted to 7.6 using concentrated hydrochloric acid (Sigma-Aldrich, Darmstadt, Germany) and the solution made up to 1 L. TBS/Tween 20 1% solution was prepared by mixing 5 mL of Tween 20 (Becton Dickinson, NJ, USA) with 500 mL of TBS.

#### 4.3.2. Antibody Titration and Immunohistochemical Staining Procedure

Antibody titration experiments were performed to establish the optimal dilution for each primary antibody (1:12.5, 1:25, 1:50, 1:100, 1:200 or 1:400). Each concentration was used for staining tissue sections for 1, 2, 4 and 24 h. The staining procedure was carried out as described elsewhere [33]. In summary, formalin-fixed, paraffin-embedded (FFPE) slides were immersed in xylene, a wax removal and antigen retrieval solution, at 96 °C for 30 min using a metal trough. After allowing the W-CAP to cool to room temperature for 20 min, samples were rinsed and rehydrated in cold water for two minutes. Excess water was removed from the slides taking care not to disturb the tissue sections or allow them to dry out, and margins were drawn around the slide sections using a wax pen (ImmEdge Pen, Vector Labs, Burlingame, CA, USA). The slides were laid flat in a humidified chamber and two drops of TBS/Tween 20 1% solution applied to each section for 5 min. The solution was tipped off and excess fluid dried from the slides with tissue paper prior to the addition of the antibodies. Based on the results of these experiments detailed below, specific dilution factors and incubation periods were used (Table 2). 

An aliquot of 100 μL of the diluted primary antibody (all from BD Pharmingen, San Diego, CA, USA) was applied to each section and the slides incubated at room temperature for either 1 h (CD3, CD8, CD20, CD45RO) or 24 h (CD4, CD69) (Table 1). For the 24-h-incubation, the container was left at room temperature on an orbital shaker to ensure even coverage of tissue section with antibody solution. At the end of the incubation the primary antibody was removed from the sections, which were then washed with TBS/Tween 20 solution for 5 min and excess fluid dried off. The slides were put back in the humidified chamber and two drops of the secondary biotinylated rabbit anti-goat IgG (DAKO, Glistrup, Denmark) antibody was applied to each section. 

The slides were then incubated at room temperature for 30 min before sections were washed with TBS/Tween 20 for 10 min followed by TBS for another 10 min. During the wash an appropriate volume of diaminobenzidine (DAB) solution (DAKO, Glistrup, Denmark) was prepared by mixing one drop of DAB concentrate in 1 mL DAB Buffer. Once washed, the excess TBS was removed from the sections and 100 μL of the diluted DAB solution applied to each section and incubated for five minutes after which the sections were checked microscopically for the correct stain level. The slides were immersed in water for a minute and counterstained with Mayer’s Haemoxylin (DAKO, Glistrup, Denmark) for 20 s before placing them in water for 5 min. The slides were then dehydrated by placing them in 99.9% Ethanol for 5 min twice and then in xylene for 5 min. A drop of distrene, plasticizer, xylene (DPX, Sigma, Darmstadt, Germany) was placed on top of the sample, which was then sealed with a coverslip. Once dry, slides were examined by light microscopy.

### 4.4. Photography and Quantification of Lymphocyte Subsets

All stained sections were systemically examined using a Phillips CM10 Transmission Electron Microscope under 40× magnification and images were captured using a side mounted Gatan BioScan camera. Sections with optimal staining intensity were selected for image analysis. A graticule fitted into one of the eye pieces of the microscope facilitated cell counting on the tissue sections. A cell was defined as positive for staining if the appropriate marker for that cell type was observed. Two sets of values were obtained, one for the DAB-stained cells and the second for the DAB- and haematoxylin-stained cells. Results were expressed as percentage of positive cells out of the total cells, as described elsewhere [34]. For spleen sections, only stained cells in the red pulp regions were counted and recorded with the counting done in approximately similar regions in all sections used. For consistency’s sake, three fields were counted per tissue slide in order to determine the presented frequencies.

### 4.5. Statistical Analyses

Means and standard deviations (except for SMA) were then calculated using GraphPad Prism version 6.01 for Windows (GraphPad Software, San Diego, CA, USA).

## Figures and Tables

**Figure 1 pathogens-11-00851-f001:**
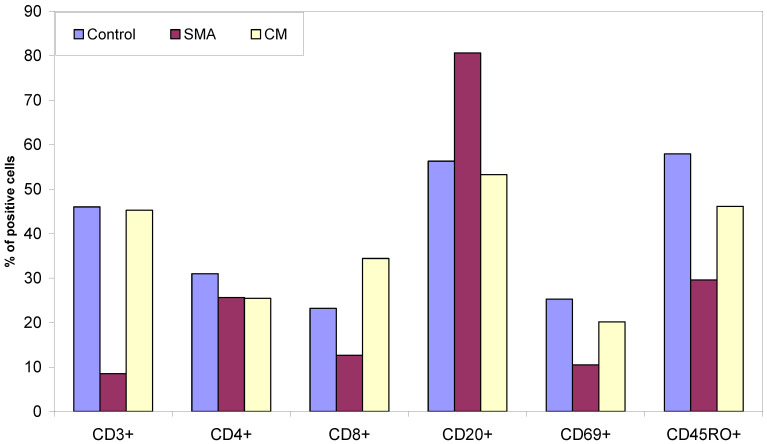
Percent of CD3^+^ T cells, CD4^+^ and CD8^+^ T cells, CD20^+^ B cells, CD69^+^ and CD45RO^+^ cells present in lymph node tissue sections from patients who died from non-malaria disease, SMA and CM.

**Figure 2 pathogens-11-00851-f002:**
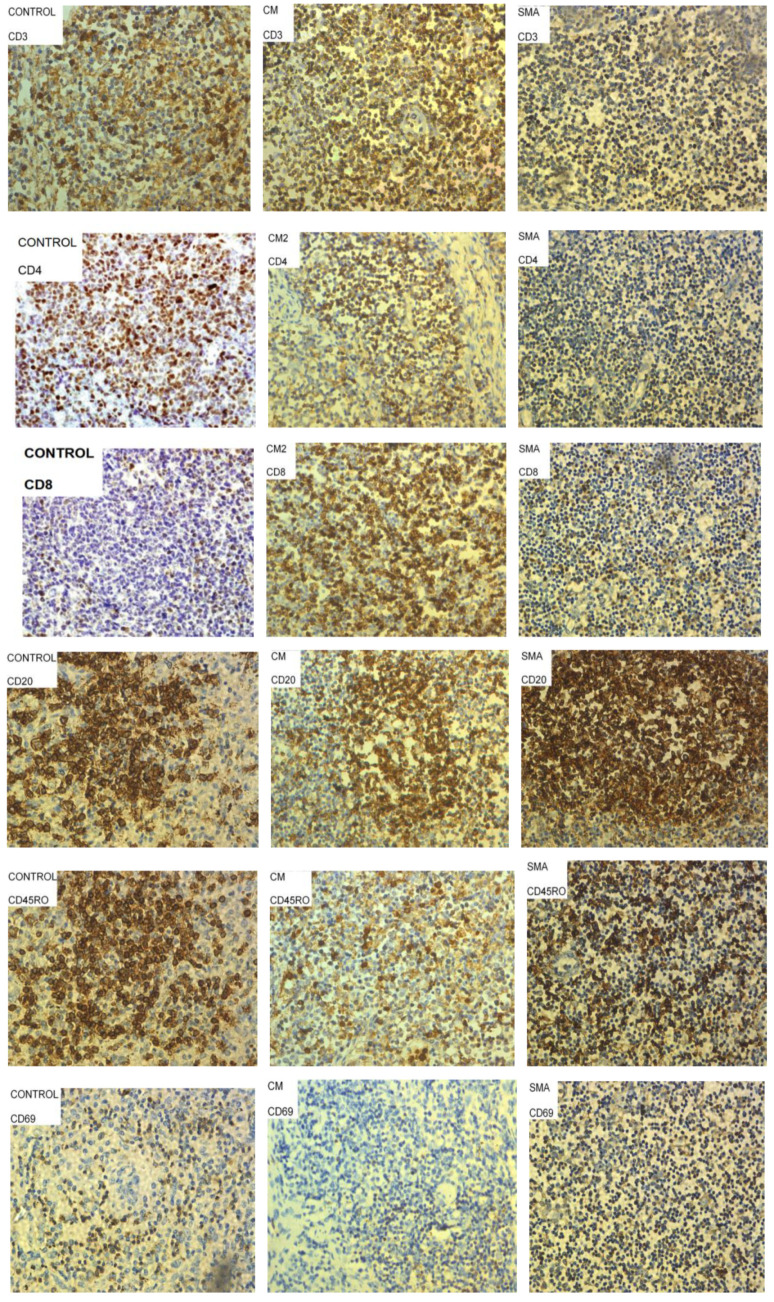
Representative images of lymph node tissue sections from patients who died of sepsis (controls), Severe Malarial Anemia (SMA) and Cerebral Malaria (CM) stained with antibodies to characterize CD4^+^ T cells, CD8^+^ T cells, CD20^+^ B cells, activated cells (CD69^+^) and memory cells (CD45RO^+^). Cells stained brown were considered positive for the marker of interest, and representative examples are circled and highlighted by arrows in some panels above.

**Figure 3 pathogens-11-00851-f003:**
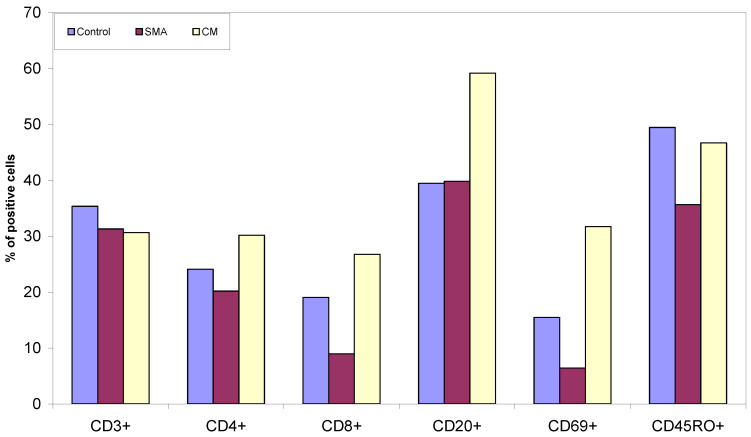
Percent CD3^+^ T cells, CD4^+^ and CD8^+^ T cells, CD20^+^ B cells, CD69^+^ and CD45RO^+^ cells present in spleen tissue sections from patients who died from sepsis, SMA and CM.

**Figure 4 pathogens-11-00851-f004:**
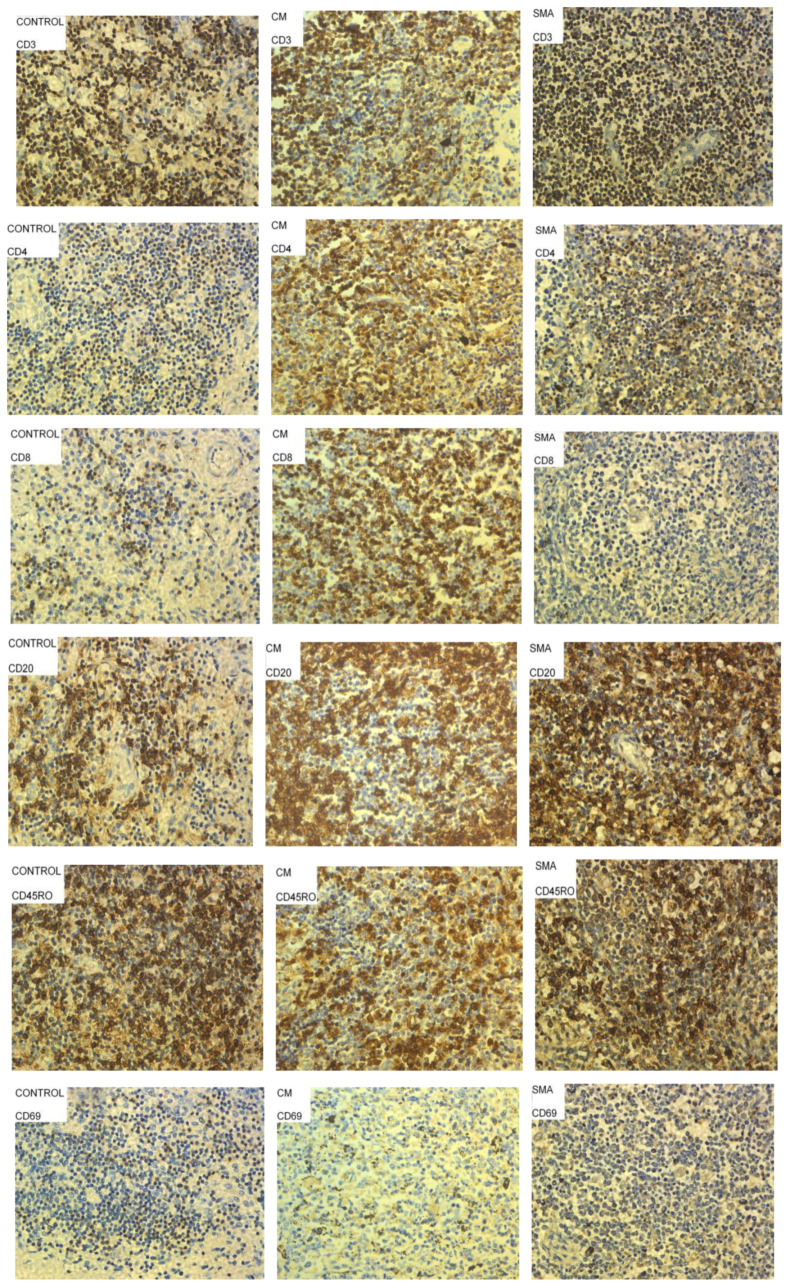
Images of spleen tissue sections from patients who died of sepsis (Controls), Severe Malarial Malaria Anemia (SMA) and Cerebral Malaria (CM) stained with different antibodies to characterize CD4^+^ T cells, CD8^+^ T cells, CD20^+^ B cells, activated cells (CD69^+^) and memory cells (CD45RO^+^). Cells stained brown were considered positive for the marker of interest, and representative examples are circled and highlighted by arrows in some panels above.

**Table 1 pathogens-11-00851-t001:** Dilution factors of the primary antibodies (BD Pharmingen, CA, USA) and the incubation duration for staining with the primary antibody and their source.

Antibody Type	Cell Types and Status	Dilution Factor of Primary Antibody	Incubation Time for Primary Antibody (Hours)
CD3	T cells	1:50	1
CD4	CD4^+^ (Helper) T cells	1:50	24
CD8	CD8^+^ (Cytotoxic) T cells	1:50	1
CD20	B cells	1:200	1
CD45RO	Memory cells	1:100	1
CD69	Activated cells	1:25	24

**Table 2 pathogens-11-00851-t002:** Diagnosis details of the patients from whom the tissue sections were collected.

Serial No.	Code Number	Diagnosis Details	Age (Months)	Sex
1	-	Control	48	Male
2	-	Control	50	Female
3	MP-55	Cerebral malaria	52	Male
4	MP-64	Cerebral malaria	50	Male
5	MP-66	Cerebral malaria	14	Female
6	MP-57	Severe Malarial Anaemia	49	Female

## Data Availability

The data presented in this study are available on request from the corresponding author. The data are not publicly available due to ethical reasons considering that the human tissue sections were obtained from postmortem samples.

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
