# Peer review of "Characterization of Lymphocyte Subsets in Lymph Node and Spleen Sections in Fatal Pediatric Malaria"

_pathogens, 2022, doi:10.3390/pathogens11080851_

Round 1
Reviewer 1 Report
Please see attached

Author Response
Reviewer 1
Summary: Dr Mandala and colleagues presented a study investigating lymphocyte subsets in lymph node in fatal paediatric malaria. They reported higher percentage of T cells in CM subjects compared with SMA subjects, and in SMA subjects, percentage of B cells were more compared with CM subjects, overall suggesting homing of cells to specific tissues during the different courses of pathology. However, confounded by the small sample size.
The discussion may benefit by having a section to discuss how the differences in immunological cells contribute to protection/pathology.
Authors’ Response: We thank the Reviewer for this comment. We have now added a paragraph in the Discussion section on page 8, (lines 230 to 243) which discusses the possible benefit of the observed difference in cell counts between CM and SMA tissue sections.
Minor
- Lines 46-59 in my opinion is not necessary. This MS investigate on the different phenotypes of immunological cells, would be more benefitting to discuss the possible pathological roles these cells play. Eg increased T cells in the spleen, what does this suggest?
Authors’ Response: We thank the Reviewer for this comment. Much as we agree with the Reviewer that this manuscript is mainly focusing on characterizing tissue section of patients who died from different clinical types of malaria, we are still of the opinion that this first paragraph in the Introduction section introduces the different clinical types of malaria and clarifies why malaria is still a disease of interest. However, we have revised this opening paragraph cutting off some segments but still conveying the necessary introduction on severe malaria and its effect on young children in SSA. Please refer to Page 1 (lines 33 to 36)
- I am not sure if line 71-72 provided the corroboration of the earlier cited mouse studies. Probably reword to make it clearer
Authors’ Response: We have now clarified this point on page 2 (line 79) of the revised manuscript.
- Lines 81-83, it is worth mentioning what is the differences in this study compared with citations 23-24, at present seems like this is a repeat on earlier studies
Authors’ Response: We have now revised the statement on page 2 (lines 92-95) to differentiate our study from the other two studies which have been cited in References number 23 and 24.
- Table 2, it would be good to provide what does each marker stained for eg, on T cells, B cells? Lines 104 to 117, do provide the percentages for easy reading
Authors’ Response: We have now revised Table 2 (which is now Table 1 due to re-arrangement of various sections in the manuscript). Please refer to page 3 (lines 112-113) of the revised manuscript
- CD69 apoptic signal
Authors’ Response: We have also indicated that CD69 in our case was used as an activation marker and not as a marker for apoptosis. Please refer to Table 1 on Page 3 (lines 112 – 113) of the revised manuscript
Major
- Resolution for fig 2 and 3 is not great. It is very difficult to quantify. Some arrows might be helpful
Authors’ Response: Indeed the Reviewer makes a valid point. The resolution for Figures 2 and 3 are not as good as we would have loved them to be. We have now included arrows as advised by the Reviewer to point out the positively stained cells.
- The study lacks apoptotic markers, eg. CD69 may be an early marker of apoptosis, and the increase in CD69 expression needs to be better characterized (line 197-199)
Authors’ Response: Yes, the study does lack a marker for apoptosis. Unfortunately, as mentioned earlier, we did NOT use CD69 in this case as a marker of apoptosis but as an activation marker.
- Line 200, P chaubaudi is not a known model of CM, is there any study done with P berghei that supports the associations with Line 208-209?
Authors’ Response: Indeed this is an important observation by the Reviewer. We have now changed Reference number 31 and replaced it with one on P. berghei. Please refer to page 8 (lines 261 to 268) of the revised manuscript and the new Ref No. 31 (Ghazanfari et al, J Immunol 2021.)
Specific
- Keep American /British words consistent, eg paediatric (if British) line 3
Authors’ Response: This has now been done throughout the revised manuscript
- Line 40 (ADRS) is not needed, not used later
Authors’ Response: We have now removed ADRS from the Introduction section. Refer to page 1 (lines 37 to 38) of the revised manuscript
- Line 157 “falciparum”
Authors’ Response: We have now made this change on page 8 (line 217) of the revised manuscript.
Reviewer 2 Report
Authors characterise in this paper the lymphocyte distribution in lymph nodes and spleens in the sections derived from post-mortem paediatric malaria patients. The samples are precious and the study is important. Some methodological and descriptive issues are mentioned below.
1. Table 1 is missing.
2. Reorganize the column sequence in Figures 1 and 3 regarding the patient pathology. This sequence must correspond to the sequence of microscope images in the Figures 2 and 4 correspondently, which are the primary data for the Figures 1 and 3. In actual draft: Control /SMA/CM in the Fig 1, 3 and Control/CM/SMA in Fig 2, 4.
3. The sequence of described markers must be the same (e.g. CD69+ is second-last in Fig 3 and the last in the Fig 2).
4. Fig 1 and Fig 3 error bars with SD are missing (even made from few samples).
5. Table 3 repeats the data from Fig 1, Fig 3. For me will be better to join Table 3 with Fig 1 and Fig 3, reporting SDs in the figures.
6. Fig 2 and Fig 4 legends: Expand the legend describing all abbreviations and signs used in the figures? E.g. what “CM2” from the 3rd row means?
7. The major result of this paper is markers’ expression on smear sections, based on DAB- and haematoxylin-stained cells’ quantification method, which is not described in the Methods, but linked to the paper [28] from the reference list. Due to the importance of the method and quite low online accessibility of [28] which is Immunology Letters from 1995, I ask to describe meticulously the staining and image reading method with attention for background details.
8. In connection with methodological question above my doubt is rise for presented results: in Fig 1 and Table 3 the percentage of CD8 positive cells are 23,2+-4.53% in control (in the Fig 2, line 3 of the microscopy images, in the control panel, for my personal criteria CD8+ cells look 10-15%). In the case of CM 34,41+-4.88% in the Fig 1 and Table 3 are reported (on the Fig 2 on CD8, CM2 image panel for my criteria roughly 60% of positive cells are reported). For CMA, 12.64% are reported in the Fig 1 and Table 3 (instead, on the Fig 2, only 3-5% for me).
In all cases a high percentage of doubtful cells are presented, thus the methodological description how these cells were managed is strongly needed.
9. The first chapter in Results “Immunohistochemistry” is more fitted to Methods part.
10. Please, add some more data about the patients, as age, which is very important for immune response develop.
Author Response
Reviewer 2
Authors characterise in this paper the lymphocyte distribution in lymph nodes and spleens in the sections derived from post-mortem paediatric malaria patients. The samples are precious and the study is important. Some methodological and descriptive issues are mentioned below.
- Table 1 is missing.
Authors’ Response: A revised Table 1 is included on page 2 and a revised Table 2 is on page 10 in the revised manuscript
- Reorganize the column sequence in Figures 1 and 3 regarding the patient pathology. This sequence must correspond to the sequence of microscope images in the Figures 2 and 4 correspondently, which are the primary data for the Figures 1 and 3. In actual draft: Control /SMA/CM in the Fig 1, 3 and Control/CM/SMA in Fig 2, 4.
Authors’ Response: This has now been done. Please refer to the revised Figure 2 (page 5) and Figure 4 (page 7).
- The sequence of described markers must be the same (e.g. CD69+ is second-last in Fig 3 and the last in the Fig 2).
Authors’ Response: This has now been done. Please refer to the revised Figures 2 and 4 (refer to pages 5 and 7 respectively)
- Fig 1 and Fig 3 error bars with SD are missing (even made from few samples).
Authors’ Response: This has now been done. Please refer to the revised Figures 1 and 3 (Refer to pages 4 and 6 respectively)
- Table 3 repeats the data from Fig 1, Fig 3. For me will be better to join Table 3 with Fig 1 and Fig 3, reporting SDs in the figures.
Authors’ Response: We have now eliminated Table 3 from the manuscript as advised. Please refer to page 3.
- Fig 2 and Fig 4 legends: Expand the legend describing all abbreviations and signs used in the figures? E.g. what “CM2” from the 3rd row means?
Authors’ Response: This has now been done. Please refer to the revised legends for Figures 2 and 4 (pages 5 and 7 respectively). We have now corrected CM2 to just CM since the number was merely indicating that the picture being displayed had been taken on tissue section from the second CM case.
- The major result of this paper is markers’ expression on smear sections, based on DAB- and haematoxylin-stained cells’ quantification method, which is not described in the Methods, but linked to the paper [28] from the reference list. Due to the importance of the method and quite low online accessibility of [28] which is Immunology Letters from 1995, I ask to describe meticulously the staining and image reading method with attention for background details.
Authors’ Response: We have now revised this section and replaced reference number 28 with a new one that clearly states how the actual staining was done. Please refer to page 11, lines 388 to 400. The new Reference 28 is now Cathro HP and Stoler MH. 2006.
- In connection with methodological question above my doubt is rise for presented results: in Fig 1 and Table 3 the percentage of CD8 positive cells are 23,2+-4.53% in control (in the Fig 2, line 3 of the microscopy images, in the control panel, for my personal criteria CD8+ cells look 10-15%). In the case of CM 34,41+-4.88% in the Fig 1 and Table 3 are reported (on the Fig 2 on CD8, CM2 image panel for my criteria roughly 60% of positive cells are reported). For CMA, 12.64% are reported in the Fig 1 and Table 3 (instead, on the Fig 2, only 3-5% for me). In all cases a high percentage of doubtful cells are presented, thus the methodological description how these cells were managed is strongly needed.
Authors’ Response: Indeed the Reviewer’s observations are fully appreciated. Actually, the manner in which the Reviewer quantifies the stained cells does not contradict our observations but in fact adds more weight to our proposed conclusion that the sequestration of both CD8+ (and CD4+) T cells is higher in CM cases compared to SMA cases since the Reviewer proposes higher percentages of stained cells. However, we have now edited the section in the Methods and Materials that describes how the staining and interpretation of the slides were done. Please refer to page 11, lines 388 to 400.
- The first chapter in Results “Immunohistochemistry” is more fitted to Methods part.
Authors’ Response: We have now edited this section and have moved the part that should go into the M&M section to the appropriate section. Refer to page 11, lines 366 to 370.
- Please, add some more data about the patients, as age, which is very important for immune response develop.
Authors’ Response: We have now provided the age and sex of the patients in the Revised Table 1 (refer to page 10).
Round 2
Reviewer 1 Report
Histology remains poor. It is difficult to identify / differentiate the different staining.
Author Response
We thank the Reviewer for the additional comments that have been raised.
- Fine/ Minor Spell Check Required
Authors' Response: We have now done a thorough speck check and the attached version is a revised version of the manuscript
2. Histology remains poor. It is difficult to identify / differentiate the different staining
Authors' Response: We have now included this point in the Limitations section on page 9. Indeed, we would have loved a better staining outcome of the tissue section unfortunately what we are presenting are the staining we obtained. You will notice in the manuscript that we refer to the Urban et al, I&I 2005 paper on Vietnamese adults as the main study that has done similar work like ours. They too had some challenges with the staining of the tissue sections. However, a recent study on mice models (Wang, et al, Malaria J 2021) which stained spleen tissue sections had much better stained slides. Its possible, the problem could be to do with using tissues sections that had been archived for a while before being stained.